# Blood Biochemical Variables Found in Lidia Cattle after Intense Exercise

**DOI:** 10.3390/ani11102866

**Published:** 2021-09-30

**Authors:** Francisco Escalera-Valente, Marta E. Alonso, Juan M. Lomillos-Pérez, Vicente R. Gaudioso-Lacasa, Angel J. Alonso, J. Ramiro González-Montaña

**Affiliations:** 1Academic Unit of Veterinary Medicine, Autonomous University of Nayarit, Tepic 69130, Mexico; fescalera@uan.edu.mx; 2Animal Production Department, Veterinary Faculty, University of León, 24071 León, Spain; marta.alonso@unileon.es (M.E.A.); v.gaudioso@unileon.es (V.R.G.-L.); 3Production and Health Animal, Public Health Veterinary and Science and Technology of Food Department, Veterinary Faculty, Cardenal Herrera-CEU University, 46115 Valencia, Spain; juan.lomillos@uchceu.es; 4Medicine, Surgery and Anatomy Veterinary Department, Veterinary Faculty, University of León, 24071 León, Spain; ajalod@unileon.es

**Keywords:** metabolic profile, *Bos taurus*, exercise, stress, fighting bull, Lidia breed

## Abstract

**Simple Summary:**

There are limited published data in cattle on blood biological variables in response to intense work or after significant physical exertion. Stress should be considered in addition to exercise in the Lidia breed due to their agonistic behavioral characteristics. Therefore, changes in blood biological variables in response to intense exercise and stress were evaluated. Specifically, 438 bulls of the Lidia breed were sampled after a bullfight in order to determine the concentrations of different blood variables. We found that most of the measured blood variables clearly increased. Thus, severe hyperglycemia and an increase in various enzymes (LDH, AST, GGT, ALT, and particularly CK) were observed. The increase in all these variables is justified by the mobilization of energy sources, tissue and muscle damage, and dehydration, due to intense exercise and stress.

**Abstract:**

There are limited published data in the bovine species on blood biological variables in response to intense work or after significant physical exertion. Lidia cattle, in addition to their exercise components, have some behavioral agonistic features that make them more susceptible to stress. The bullfight involves stress and exercise so intense that it causes significant changes in some metabolic variables. The study objective was to evaluate changes in blood biological variables in response to intense exercise and stress. After the fight in the arena, and once the bulls were dead (n = 438), blood samples were taken, and some biochemical and hormonal variables were determined in venous blood. A descriptive analysis was performed using the Statistica 8.0. computer program. The mean (±s.d.) results obtained were: total protein (85.8 ± 10.8 g/dL), albumin (3.74 ± 4.3 g/dL), triglycerides (39.65 ± 0.16 mg/dL), cholesterol (2.44 ± 0.03 mmol/L), glucose (22.2 ± 9.6 mmol/L), uric acid (340 ± 80 µmol/L), creatinine (236.9 ± 0.4 µmol/L), urea (5.93 ± 1.27 mmol/L), LDH (2828 ± 1975 IU/L), CK (6729 ± 10,931 IU/L), AST (495 ± 462 IU/L), ALP (90 ± 33 IU/L), GGT (50 ± 34 IU/L), ALT (59 ± 35 IU/L), cortisol (117.5 ± 46.6 nmol/L), and testosterone (20.2 ± 23.8 nmol/L). Most of the measured variables clearly increased; thus, we found severe hyperglycemia and increases in LDH, AST, GGT, and ALT enzymes, particularly in CK. The increases in all these variables are justified by the mobilization of energy sources, tissue/muscle damage, and dehydration due to continued stress and intense exercise.

## 1. Introduction

Work and strenuous physical exercise involve the activation of several metabolic pathways when trying to cope with these demanding situations. Most studies on exercise physiology have, in addition to human studies, been made in horses [1,2,3,4,5,6,7,8,9,10], dogs [11,12,13], and camels [14], as well as species involved in various types of competitions by what have been called “exercised species” [15].

There are a few studies in cattle that have addressed the impact of performing intense exercise [16,17,18,19,20], and a few studies have measured blood variables after periods of intense work and physical exhaustion. However, there are more publications on blood changes found in cattle under stressful situations such as livestock transport [21,22,23], food deprivation [22,23,24,25,26], heat stress [27,28], and some livestock diseases [29,30].

Lidia cattle (*Bos Taurus brachiceros*) [31], the second pure breed in the bovine census in Spain, are a heterogeneous Iberian cattle population, described in Decree 60/2001 [32], which are mainly used for breeding under extensive conditions in central and southern Spain, France, Portugal, and Latin America countries [33,34]. This breed is known for its natural aggressiveness and resistance to traditional handling procedures [31,33]. The cattle have access to grass in the “*dehesa*” landscape and are usually supplemented with fodder or concentrates in the summer or during the winter, if there is a shortage of pasture. In the year before the bullfight, the bulls are supplemented with concentrates during a period commonly known as “finished” [18,33]. Other characteristics of their handling, feeding, and aptitudes are extensively described in Lomillos and Alonso [33].

The bulls of this breed, 4–5 years old, are transported from the livestock to the arena where the fight takes place (also called “*lidia*”). They remain for a few hours or a maximum of a few days in the “*corrales*” (where they are placed individually), until their departure to the fighting and ends with the death of the bull. During the fight, these animals undergo intense exercise, which lasts between 15 and 20 min. During this period, the animal, usually with some previous training, performs intense athletic activity [15] that combines periods of intense exercise followed by rest periods of variable duration. The high-intensity exercise with anaerobic metabolism, which generates a significant amount of lactate [18], reaches its maximum energy yield in about 30 s, while periods of low demand use aerobic or slow oxidative metabolism [15]. While in the fight, the rest periods serve the bull to regain its strength [15]. This type of exercise can be likened to intermittent exercises performed by standard breed horses with intermittent training [35].

Exercise not only induces oxidative stress, but can also modify the physiological animal metabolism, leading to changes in hematological and biochemical variables [8,35,36,37,38,39,40]. Knowledge of metabolic values is essential to monitor the normal homeostatic functioning of the organism.

Glucose is the most widely used indicator of energy status. As glucose is the major metabolite of respiratory oxidation, lactose production by hepatic metabolism is vital for brain metabolism [41]. Fasting, excessive and/or prolonged exercise, excessive insulin administration, and certain liver lesions (also hormonal imbalances) can induce significant changes in blood glucose. Proteins are involved in the transport of multiple substances, and in maintaining oncotic pressure and humoral immunity [41], and their plasmatic concentration is modified by multiple factors such as age, growing, hormone profile, sex, pregnancy, and lactation, as well as nutritional status, stress, and loss of fluids [42]. The values of some metabolic parameters such as triglycerides, uric acid, creatinine, and urea allow us to check for correct global body functioning. These latter metabolites increase due to kidney disorders, although other circumstances such as diet, significant physical efforts, age, or sex may cause an increase in blood values [41]. The activity of some enzymes have clinical significance when values are outside the normal reference ranges [43]. For example, lactate dehydrogenase (LDH) and creatine kinase (CK) increase in cell injury such as myocardial infarction, liver and kidney disease, tumor processes, and cerebrovascular disease [41]. Alkaline phosphatase (ALP) is important for the transport of sugars and phosphates in the intestinal mucosa, renal tubules, bone, and placenta, keeping in mind that all organic cells using glucose for energy require a phosphatase [41].

It has been reported that there are significant blood changes in Lidia cattle as a result of the fighting [16,17,18,20]. However, these changes have not been examined in a larger population of bulls using multiple measurements of energy metabolism, nor are there enough justifications cited to explain these metabolic modifications in bulls.

Therefore, the aims of our study were, on the one hand, to investigate metabolic changes that occur in bulls as a result of the fighting, where short periods of intense exercise are combined with intermittent periods of rest and where the exceptional conditions of this exercise must be especially considered; and, on the other hand, to try to justify why these modifications occur, even resorting to arguments used in other species, such as horses, dogs, or even humans, subjected to similar situations of exercise and stress.

## 2. Material and Methods

Animals. A total of 438 four- to five-year-old male Lidia bulls from different farms, fought in León, Burgos, Valladolid, and Salamanca bullrings located in the Region of Castilla and León, Spain, were used in the present study. Before the bullfight, the bulls were inspected by the Official Veterinary Services of each bullring to check their suitability for the fight, and to ensure that they were in good health. After the bullfight, all the animals were slaughtered under local regulation law [44].

The bulls were transported from the breeding farms to the bullring (arena location), always in accordance with the legislation on transport and animal welfare [45] in the days prior to the bullfight. Water and feed are provided there until a few hours before the fight, although they can be maintained if the weather is very hot. The fight of each bull lasts around 15 min.

Ethics statement. All Lidia bulls, from different breeding farms used in this study, were fought in different arena locations under the regulation of the local legislation law [44], and at the moment of the sampling, all animals were already dead. In this way, we did not impose any additional experimental procedures that should cause any suffering or pain. All experimental procedures were performed in compliance with the provisions of the EU Directive regulating the use of animals for scientific purposes [46] and the Decree that regulates experimentation and animal protection in Spain [47].

Samplings and biochemical analysis. The blood samples were collected into heparinized vacuum tubes (9 mL) immediately after the animal died. The samples were centrifuged at 4000 rpm (2200× *g*) for 10 min (following the methodology described by Escalera-Valente et al. [18]). Following centrifugation, the plasma was collected free of impurities, placed in Eppendorf tubes, stored at 4 °C for up to 3 h, and subsequently stored at −20 °C until analyzed in the Laboratorio de Técnicas Instrumentales (L.T.I.) in the University of León. To determine some biochemical variables, a Cobas Integra 400 (Roche) multi-analyte analyzer was used, by means of Roche Diagnostic reagents. The biochemical variables measured were glucose, total proteins, albumin, triglycerides, cholesterol, creatinine, uric acid, urea, and the enzymes aspartate aminotransferase (AST), alanine aminotransferase (ALT), alkaline phosphatase (ALP), gamma glutamyltransferase (GGT), creatine kinase (CK), and lactate dehydrogenase (LDH). Plasma cortisol and testosterone concentrations were determined with an automated analyzer system (Immulite 2000, Siemens Medical Solutions Diagnostics, Los Angeles, CA, USA) and reagents were supplied by Siemens Health Care Diagnostics Products Ltd., Llanberis, Gwynedd LL55, 4EL, UK. Cortisol was determined using a solid-phase competitive chemiluminescent enzyme immunoassay. The reported calibration range for the assay was 28 to 1380 nmol/L with an analytical sensitivity of 5.5 nmol/L. The intra-assay and inter-assay percent coefficients of variation were calculated using three different plasma concentrations that ranged between 5.8 and 8.8%, and 6.3 and 10.0%, respectively. Testosterone concentration was measured by an automatic two-site sandwich immunoassay with chemiluminescent detection. Assay sensitivity was 15 ng/dL (0.5 nmol/L), the calibration range was 20–1600 ng/dL (0.7–55 nmol/L), and the intra- and inter-assay coefficients of variation were 6.8–13.0% and 7.7–16.4%, respectively.

Statistical analysis. Plasma concentrations of each parameter in blood plasma were calculated. The data were analyzed using the Statistica 8.0 statistical software. Descriptive statistics were performed, indicating the mean value, the standard deviation (SD), the mean standard error (SE), and the minimum and maximum ranges.

## 3. Results

Plasma protein, albumin, lipids, cholesterol, and glucose concentrations are shown in Table 1. The plasma concentrations of uric acid, creatinine, urea, cortisol, and testosterone are listed in Table 2. Plasma glucose concentrations are represented in Figure 1. Finally, the plasma concentrations obtained from the enzymatic and hormonal profile are listed in Table 2 and Table 3.

The most striking results were perhaps obtained in blood glucose, with high values. Thus, in some bulls, we found concentrations of 59.1 mmol/L, exceeding by more than twice the mean value of all sampled animals (22.2 ± 9.6 mmol/L) (Figure 1; Table 1), with the values being found greater than those indicated as a reference for this species [41,48,49]. Mean (±SD) plasma cortisol concentrations were 117.5 nmol/L (±46.6), with a range from 27.59 to 386.26 nmol/L. Mean plasma testosterone concentrations were 20.20 nmol/L (±23.86), with values ranging from 2.04 to 104.10 nmol/L.

In general, the enzyme profile showed activity values above the usual ones for cattle, with CK, AST, LDH, and GGT showing the most marked increases (Table 3). Interestingly, plasma CK activity was 1000 fold greater than the values reported for the normal physiological range for cattle, and even the CK activities were 93,000 IU/L in one animal and 6800 IU/L in several animals (Figure 2).

## 4. Discussion

Clinical chemistry analysis provides an objective means of assessing the nutritional status of an animal, but it can also be used to identify other global and specific problems. In Table 1, Table 2, and Table 3, we have included, in addition to the values found in the sampled bulls, the reference values of different parameters, both in domestic breeds and in Lydia cattle, to be compared with our values. Of course, it would be much more interesting to compare them with values obtained by sampling the bulls prior to the fight, and even before leaving the breeding farms. However, this is not possible. The breeders do not allow it, for several reasons: the bulls could be injured in the sampling, losing all their economic value, and their behavior could also be altered during the fight. Let us not forget the difficult handling and the special behavior of this breed.

Activation of the hypothalamic pituitary adrenal (HPA) and sympathetic adrenal medullary (SAM) axes in response to stress prepares the animal to cope metabolically with some external stimuli, generally in the form of altered protein and energy metabolism. Thus, examination of these biomarkers provides valuable information on the status of the animal under physical overexertion and/or stress. Initiating and maintaining the stress response are expensive in terms of energy, and in order to cover energy expenditure, a continuous supply of energy is required. This supply is facilitated by anaerobic glycolysis and the aerobic breakdown of the main energy substrates, glucose and free fatty acids. Activation of the HPA and SAM axes profoundly affect the levels of these substrates, as the animal prepares to cope metabolically with stimuli. Social stress, achieved by the mixing of unfamiliar bulls in pens, resulted in elevated concentrations of glucose and NEFA [58]. In contrast, Gupta et al. reported that the concentration of glucose increased after the initial mixing of unfamiliar steers; however, no changes in the concentration of NEFA were observed [59]. Similarly, unaltered concentrations of NEFA were reported following the mixing of beef cattle [60] and sheep [61]. The findings of increased energy metabolism have generally been reported following transportation in cattle [22,58,62,63]. Reports on changes in the concentration of blood beta-hydroxybutyrate, a ketone body, are variable, with decreases [22,63,64] and increases [62,65,66] reported following transportation in cattle. 

Additionally, Sporer et al. highlighted the use of metabolites as potential biomarkers of stress following transportation in beef bulls [21]. Physical exercise, especially when it is intense, not only induces oxidative stress but can also modify the animal’s physiological metabolism, leading to changes in physiological, hematological, and biochemical variables [8,35,36,37,38]. The metabolic profiles described for Lidia cattle are similar to those described for other domestic cattle breeds [42], although some serum biochemical variables of this breed differ significantly from other cattle breeds, due to the particular temperament characteristics of this breed [17,20,42].

The plasma concentrations of total protein and albumin obtained in these animals are slightly greater than those reported for other bovine breeds [19,41,67,68,69]. However, increases in the plasma concentrations, with respect to the usual values of cattle, of both total protein and albumin, are not as marked as other variables measured in this study. Anyway, the effect of the intense physical exercise performed by Lidia cattle is still noticeable. Even when compared with those values obtained in the Lidia breed, our values are slightly greater than those cited by Jordán et al. [42], and much greater than the basal value obtained following the methodology described by Sánchez et al. [20]. 

Our results are consistent with those presented by authors such as Sporer et al. (2008) [21] as stress has effects on protein metabolism and induces significant changes in plasma concentrations of albumin, globulin, and total protein. Transport stress has been reported to increase some of these metabolites following an 8 h road journey [23,58]. The extent of the changes in metabolites is dependent on factors such as the duration of the trip, or if animals had access to feed and water during transport. Increases in total protein blood values in Lidia cattle after transport for half an hour were also reported, as well as after a restraint and open-field test [20], but they were not as intense as after the fight [17], indicating that stress alone is not sufficient to affect protein metabolism. In several studies in both horses [1,7,9] and cattle [22,23,70], it has been reported that performing some kind of exercise and/or a stressful situation can increase the plasma protein concentration. Judson et al. [7] and Aguilera-Tejero et al. [1] in horses and Carpintero et al. [71] and Parker et al. [70] in bulls stated that increased total proteinemia values are due to dehydration during transport, exercise, or fasting, whereas our results show that protein mobilization, to offset energy demand, could also contribute to the significant hemoconcentration due mainly to intense exercise and fluid loss by sweating and bleeding. On the other hand, cattle with different pathological processes affected by metabolic acidosis showed greater plasma protein concentrations [68], which is consistent with our findings as these bulls, as a result of oxidative metabolism, showed significant metabolic acidosis [18].

Plasma albumin concentrations are similar to those reported by most references in different breeds [21,22,41,48,72], and even similar to those indicated by Bartolomé et al. (2005) [54] in bullfighting bulls. Transport stress in Aberdeen Angus, Friesian, or Belgian Blue bulls is capable of reducing both total protein (11%) and albumin concentrations (7%) [21]. Increases in both serum protein and albumin levels, according to several researchers, are caused by dehydration, due to fasting, stress, and/or exercise [1,7,70], and in both cattle and horses, though some studies have reported minimal changes [73]. When we compared our albumin blood data with values reported in cows with fatty liver degeneration, we found similar plasma concentrations in medium-intensity cases and greater and smaller values when compared with mild and severe hepatic steatosis, respectively [74].

In our research, plasma triglyceride concentrations are greater than those reported in other breeds of cattle [41,48,49], as well as basal values in Lidia cattle [20], being much lower than the values cited by Bartolomé et al. in the bulls fought [54]. Plasma cholesterol concentrations are in range of those of Kaneko et al. [41]. The mean value reported is numerically greater than those reported by Alonso et al. [17] in bulls after the fight.

Although several publications mentioned increases in plasma triglyceride concentrations after exercise [75,76,77,78], others pointed to decreases [79,80] or showed no significant differences [81,82]. This could possibly be attributed to intensity and training duration [78,83] and the recovery period elapsed after exercise [77,79]. Our results are in accordance with these authors because the blood samples were taken immediately after the end of the fight and, at this moment, the triglyceride blood values were at their maximum peak considering the intensity of the exercise, without having a recovery period for the animal [77].

In the present study, plasma cholesterol concentrations are much greater than those reported in the bibliography, including baseline values measured in Lydia bulls [17,20], which seems to indicate that they increased in bulls after the fight. There are no data in the literature that compared the blood values in cattle that underwent significant exercise plus some stress. Therefore, we were forced to compare our results with the values reported in horses and humans.

In horses under stress of transport situations, plasma cholesterol concentrations are reported to be increased at the moment of unloading and up to two hours after [83]. Lennon et al. [84] described blood cholesterol plasma concentrations to increase during moderate-intensity exercise, decreasing rapidly after its cessation, but only significant in the group of moderately trained men, and also taking place in amateur cyclists compared with professional ones [75]. According to Lennon et al. [84], in humans, increased blood cholesterol was observed from 20 min of exercise, whereas for Cullinane et al., cholesterol was increased significantly only after workouts lasting two hours [79]. However, a decrease in cholesterol blood value has also been proven, which is evident from 24 h post-workout [81], and this reduction continued for three days [81]. Petibois and Déléris conducted a study for 47 weeks of training, and reported that from 15 weeks, the plasma cholesterol started to decrease [80]. 

The fight must be considered as intense exercise, so the rise in cholesterol value is consistent with previously reported studies, yet only one of the measured sample values is in the upper limit. One possible explanation is that some cholesterol is diverted to the synthesis of cortisol, a parameter that clearly increased in fighting livestock for their special features and even more so in the stressful situation these bulls present before and during the fight [17].

The plasma uric acid concentrations reported in the present study are greater than those reported for other breeds [41] and for the Lidia breed [20]. However, similar to those reported in Lidia bulls after a fight [17,54] and other stressful handling situations (Sánchez et al., 1996), there is no consensus about the influence of exercise on uric acid values. Some studies have reported that moderate exercise, in trained athletes, apparently does not affect the plasma concentrations of uric acid [85], but increased plasma concentrations of uric acid immediately after exercise have also been reported in humans [86,87].

The average creatinine plasma concentration observed in Lidia cattle after intense exercise is clearly greater than that reported in tame cattle [41,69], but it is similar to that recorded from Lidia cows when submitted to different stressful handling situations [20], which in turn is considerably greater than the values considered by these authors as the baseline of this breed. 

Our results are in accordance with Rose et al. [9] who reported that in horses, the increase in post-exercise creatinine is mainly due to the splitting of phosphocreatine in muscle, and they added that the plasma concentrations may be even greater in exercise over longer distances. In this way, the increase in blood creatinine found in our study would be due to the intense physical exercise displayed by the bull during the fight. However, stress, characteristic of this breed, given his irritability and agonistic behavior, seems to also affect creatinine blood values, which is opposite to results from a study in horses [73], where transport stress did not change creatinine values significantly.

In the present study, plasma urea concentrations are similar to reference cattle values [41,49,51] and basal Lidia values [20], although smaller than the reported values after the fight in this breed [53,54].

Bayly stated that physical activity in horses caused an elevation in uremia, especially in endurance [88]. He argued that this increase is due to an increase in protein catabolism during the gluconeogenesis process, with subsequent deamination of amino acids. Moreover, according to this author, renal perfusion is reduced in trying to restore hemoconcentration due to fluids loss. The accumulation of urea in the bloodstream persists and remains evident in the 24 h following the development of the activity [88,89]. In the same way, considering the degree of stress and exercise carried out by the animals of the present study, and their need to adapt their metabolism in that short period of time (not exceeding 20 min), they could be seen as understandable increases in uric acid, creatinine, and urea, as they are residues from nucleic acid catabolism (adenine and guanine), phosphocreatine (mainly muscle), and protein, respectively. It should also be considered that the kidney tends to retain more water, causing the hemoconcentration of those variables [9,12,18,86] because of the important physical exercise, intense sweating, and blood loss.

Blood glucose plasma concentrations in this study are significantly greater than in intensively reared domestic cattle [41,42,69], due to the greater secretion of cortisol in response to stress, typical of the breed, which is linked with temperament and excitability features of these animals [17,20,42]. In the present study, the mean plasma glucose concentrations are sevenfold greater than the values reported by Kaneko et al. [41], taking into consideration that greater values could be expected in adult males [42].

Increased plasma glucose concentrations have been reported after physical exercise in different species, including humans [85,90,91], horses [9,10,73], and dogs [11,12,13,39], which may be a consequence of the effect of the stress hormone, cortisol, on liver metabolism promoting gluconeogenesis, thus producing greater plasma concentrations in the circulation. Rose et al. suggested that the speed and duration of exercise directly influence plasma glucose concentrations [9].

In the present study, intense hyperglycemia is greater than that cited by Alonso et al. after the fight [17] and is sevenfold greater than Lidia basal values published by Sánchez et al. [20]. The fight should be considered a highly stressful activity, as plasma glucose concentrations are two-fold greater than those reported for Lidia cows that are subjected to different handling and experimental stressful situations combining immobilization, open-field tests and transport, and elevated plasma cortisol concentrations [20], correlated with the duration and complexity of the manipulations [20]. In agreement, Kenny and Tarrant reported that raised plasma glucose concentrations are indicative of the stress response, including the intensity and/or duration of the stressor [92]. In the case of fighting bulls, the stress stimulus could be affecting the animals during 2 to 3 days, commencing when bulls were transported for several hours in a truck, in a dark and confined space, and unloaded after arrival into an unfamiliar place when moving from the open spaces of the pasture to the bullring corrals.

Therefore, in our view, intense hyperglycemia is not only due to stress, but also to intense exercise, which has led to the mobilization of various sources of energy, as was stated in previous variables presented above, and although there is a significant demand of body energy, glucose cannot be metabolized. This is due, among other factors, to hypoxia, low values of oxygen partial pressure, and high carbon dioxide plasma concentrations [18]. It should also be considered that intense dehydration caused by water loss through sweating, urination, and bleeding may have induced hyperglycemia by means of hemoconcentration, reaching concentrations as great as 22.2 (±9.6) mmol/L, a maximum of 58.7 mmol/L, and several samples above 26.4 mmol/L.

Among all blood enzymes measured, although several of them clearly increased (LDH, CK, AST, GGT, and ALT), it is noteworthy that CK activities were markedly elevated in these bulls, with clearly greater activities than those reported by Kaneko et al. [41] and Merck [69] for cattle, with some bulls having 1000-fold increases in CK activity. However, the ALP blood level is within the above benchmarks in cattle [41,69]. The mean GGT activity that we have found is much lower than those reported by Alonso et al. [17] and Bartolomé et al. [54]. A possible explanation is that the bulls sampled by these researchers corresponded to a group of bulls that were not trained. As we have indicated above, until a few years ago (10–15 years), bulls were not trained, whereas today, they are trained in all (or almost all) breeding farms. In the trained animals, the metabolic overexertion will be much lower, and this is reflected in lower values of the enzymatic activity of GGT [17,54].

When comparing our values, we observed that the CK activity was greater (up to 10-fold) than basal values for this breed and also greater than values for cows subjected to the combined stressors of transportation, immobilization, and an open-field test [20], being even twice the values found by Alonso et al. [17] in bulls under the same situation. The differences found in the CK activity of bulls after the fight could be due to the on-farm physical training the bulls are subjected to nowadays but were never subjected to during the 1990s [93]. Conversely, the activities of AST, ALT, and FA are similar to those reported by Alonso et al. [17], but well above values found by Sánchez et al. [20] in these breed cows at rest or under stress. Increases in LDH, CK, and AST enzymes, and even ALP, obtained in the present study can be justified by the muscle injury and tissue destruction, including cardiac, which intense physical exercise involves [12,91,94], as muscle is one of the tissues where these enzymes are found in large quantities. Our results are in accordance with those presented in other species. In humans, post-marathon and ultra-marathon, increments in LDH, CK, AST, ALT, ALP, and GGT blood plasma concentrations have been reported [91,95]. In sled dogs after strenuous exercise, McKenzie et al. found increases in CK, AST, and ALP activities [12], and also in horses, Arias et al. found increases in the activities of CK and AST, although increases in the latter enzyme were not significant [94].

The exercise the animals perform during transportation to maintain their balance, both in horses and cattle, seems to be responsible for the increase in CK blood activity as other enzymes values, ALT, AST, and ALP, are not significantly affected [73]. Elevated CK activity in cattle is a useful measure of muscle damage and physical stress [21]. We believe that all of the abovementioned stressors, in combination with intense physical effort, induce the greatest increase in these enzymes, as only stress, even maintained over time, while producing an increase in these enzymes, is not as marked as we have found [20].

GGT is a specific enzyme for liver function in ruminants, and an indicator of hepatobiliary disorders, cholestasis, or metabolic disorders in ruminants. GGT is probably a better indicator of liver disease than ALP in ruminants [41,48]. On the other hand and although muscle damage suffered by the bull after the fight is very important, increased GGT activity cannot be justified solely by these injuries, because, according to Kaneko et al., GGT is not present in the muscle cells [41]. In our opinion and according to Fallon et al. [91] and McKenzie et al. [12], over-exertion could affect all organs, including the liver, increasing plasma activities of GGT, AST, ALT, and other enzymes, but may not necessarily be due to pre-existing liver disease. Furthermore, although increases in the values of these enzymes are considered a very sensitive indicator of hepatobiliary diseases and cholestasis in large animals, the existence of these pathologies could only be subclinical, as bulls showed no pathological clinical signs during the inspection by the Official Veterinary Services prior to the fight. The concentrate diet that [17] the bulls are fed in the year before the fight could induce a subclinical fatty liver stage that could be visible at the enzymes level when the increase in energy supply demand is such that liver metabolism is increased.

Plasma cortisol concentration, a stress hormone, which can represent between 79 and 90% of total circulating blood corticosteroids, is secreted under stressful conditions within 2–3 min after the stressor stimulus appeared, generating a response to stress situations or exercise, stimulating carbohydrate, fats, and proteins metabolism, as well as maintaining hydro-electrolyte balance [96,97].

Stressors, appearing before or during the fight, induce activation of the pituitary and adrenal sympathetic system, and ACTH is secreted, which in turn causes the rapid increase in circulating cortisol plasma concentrations, allowing us to use this hormone in the evaluation of stress similar to other cattle breeds [8,16,21,96,98,99,100,101].

Plasma cortisol concentrations reported in this study are greater than those reported for other cattle breeds [41,49,64,102], but are lower than those listed for Gomez [103] and Coppo et al. [104], although the maximum value found by us (14.8 mg/dL) is almost twice the average value described by them. Our results are in accordance with those found by Alonso et al. [17] and Chaves et al. [105], lower than those cited by Bartolomé et al. [54] and greater than those found by González-Buitrago et al. [106]

Increases in circulating plasma cortisol concentrations are a hallmark of stress in livestock [21,98,101,107], especially after the stress of transport [21,108]. As was previously mentioned, the handling and transport prior to the fight could be considered as stress stimuli. According to Sporer et al., transport greatly increases the level of blood cortisol when compared to baseline obtained 24 h before, increasing up to 528% at 4.5 h and returning to basal plasma concentrations at 24 and 48 h after transportation [21]. Taking this into account, the animals of our study would have time to recover from transport stress, but handling prior to the fight could still affect our results according to Sanchez et al. [20].

It should also be considered that physical exercise produces a significant release of cortisol, because during exercise, both the sympathetic nervous system and the hypothalamic–pituitary–adrenal axis are activated [8], which has been demonstrated in horses [8,39,40,100,109,110].

It has been reported that there are important differences between breeds, and *Bos indicus* calves have greater baseline plasma cortisol concentrations [111] than *Bos taurus* calves do, which may significantly predispose some breeds to stress [21,96,112,113]. On the other hand, in cattle, temperament differences have been linked to the stress response, with excitable (temperamental) animals having a greater baseline cortisol [96], and this could also explain the high cortisol response to the fight stressor observed in Lidia bulls.

The mean plasma testosterone concentrations reported in the present study are greater than those reported by Marcus and Durnford in cattle [114] but lower than those found by other groups [21] with the greatest concentrations being reported by Kilroy and Dobson of 3000 ng/dL [115].

Transport stress has been reported to decrease plasma testosterone to such an extent that after the 4.5 h journey started, the value was 74% lower than baseline when bulls were sampled 24 h before the start of the stimulus [21]. There is no consensus on how exercise influences the level of testosterone in horses, because no changes were reported from Estrada and Ruiz [116], while a significant increase remaining for at least 30 min post-exercise was stated by Geor et al. [117] and Lemazurier et al. [118].

In the Lidia breed, it was found that there is an increase in testosterone plasma concentrations after the fight [119,120], although the values that we found (582.2 ng/dL) are below those reported by these authors. In our opinion and according to Dunlop and Malbert [121] and Sporer et al. [21], stress causes a generalized suppression of the hypothalamic–pituitary–gonadal axis, so in bulls with increased endogenous steroids in response to stress, it is associated with reduced plasma concentrations.

## 5. Conclusions

The conclusions of the present study are limited due to the impossibility to have data of the different variables studied before the sacrifice of the animals. As a result of the stress and intense physical effort experienced by bulls during the fight and compared to values considered as “physiological” in cattle, significant increases in the values of total proteins, albumin, triglycerides, cholesterol, uric acid, creatinine, urea, and glucose, and LDH, CK, AST, ALP, GGT, and ALT enzymes and cortisol were found. These increases could be justified by the mobilization of energy sources, tissue/muscle damage, and dehydration due to continued stress and intense exercise.

## Figures and Tables

**Figure 1 animals-11-02866-f001:**
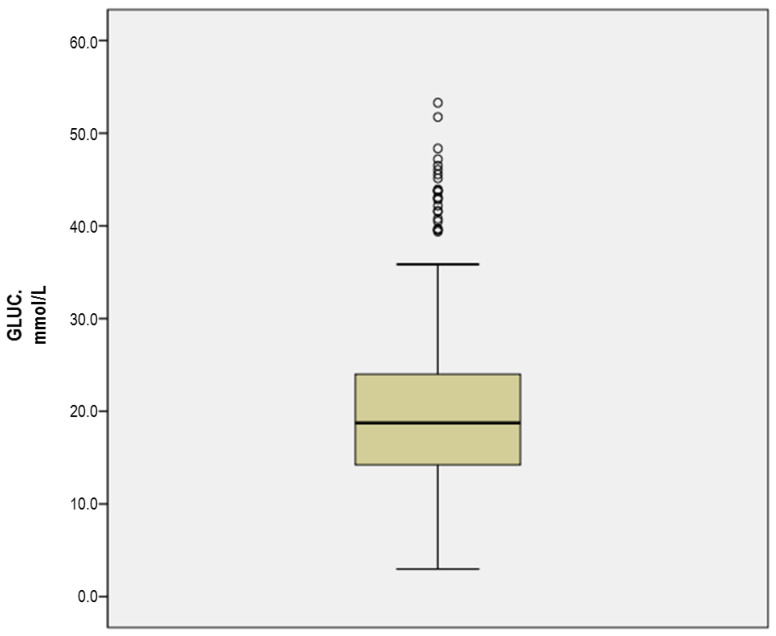
Graphic representation of the glycemia (mmol/L) of bulls after physical exercise.

**Figure 2 animals-11-02866-f002:**
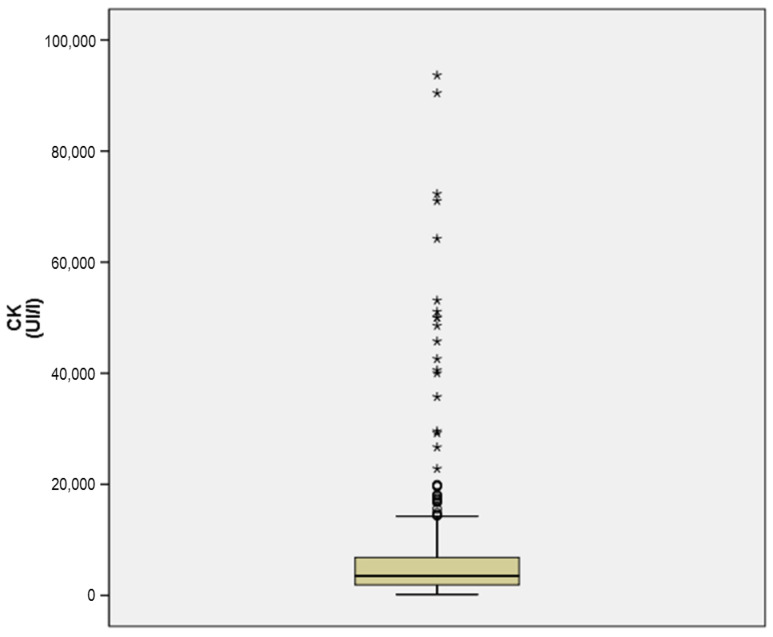
Graphic representation of the CK activity (UI/L) of bulls after physical exercise.

**Table 1 animals-11-02866-t001:** Summary statistics of biochemical variables (I) in Lidia bulls’ plasma. Values found by other authors are also shown, both in Lidia breed and in other breeds.

	Total Protein	Albumin	Triglycerides	Cholesterol	Glucose
	(g/L)	(g/L)	(mmol/L)	(mmol/L)	(mmol/L)
n	438	438	438	438	24.1
Mean	85.8	37.4	0.45	2.44	22.2
Geometric Mean	85.1	37.1	0.42	2.36	20.2
Harmonic Mean	84.5	36.9	0.39	2.27	18.2
Sum	37,580.0	16,380.0	196.1	1068.0	9733.1
Minimum	38.5	20.0	0.11	1.01	3.3
Maximum	147.8	48.5	1.17	5.30	58.7
Lower Quartile	79.4	34.5	0.34	2.00	15.6
Upper Quartile	91.5	40.2	0.54	2.86	26.4
Standard Deviation	10.8	4.3	0.16	0.65	9.6
González et al., 2000 [50] ^a^	66–90	29–41	-	3.0–5.0	2.5–4.1
Earley et al., 2006 [22] ^a^	81.3 ± 0.92	37.6 ± 0.03	-	-	5.0 ± 0.07
Radostits et al., 2006 [49] ^a^	57–81	21–36	0–0.2	1.0–5.6	2.5–4.2
Kaneko et al., 2008 [41] ^a^	67.4–74.6	30.3–35.5	0–0.2	2.07–3.11	2.5–4.1
Buckham et al., 2008 [21] ^a^	75.3 ± 0.92	28.2 ± 0.25	-	-	-
Otter, 2013 [51] ^a^	61–81	27–39			
Constable et al., 2017 [48] ^a^	57–81	21–36	0–0.2	1.0–5.6	2.5–4.2
Sánchez et al., 1996 [20] ^b^	74.0 ± 6.8	-	0.29 ± 0.06	-	3.50 ± 0.42
Alonso et al., 1997 [17] ^b^	91.8	-	0.39	1.0	20.5
Castro et al., 1997 [52] ^b^	73.0–84.3		0.20–0.29	2.9–6.7	63.1–236.3
Méndez et al., 2003 [53] ^b^	-	24	-	-	8.3
Bartolomé et al., 2005 [54] ^b^	79.8	39.7	2.07	5.0	9.3

^a^ In different breeds. ^b^ Everything in Lidia cattle.

**Table 2 animals-11-02866-t002:** Summary statistics of biochemical variables (II) in Lidia bulls’ plasma. Values found by other authors are also shown, both in Lidia breed and in other breeds.

	Uric Acid	Creatinine	Urea	Cortisol	Testosterone
	(μmol/L)	(μmol/L)	(mmol/L)	(nmol/L)	(nmol/L)
n	393	438	438	326	326
Mean	340	236.91	5.93	117.5	20.20
Geometric Mean	328	2.65	5.79	107.33	13.68
Harmonic Mean	301	2.61	5.65	93.81	10.56
Sum	133,692	1174.0	2595.4	38,294.9	6585.9
Minimum	27	0.97	2.73	27.59	2.04
Maximum	637	4.43	11.12	386.26	104.10
Lower Quartile	290	2.43	5.03	102.36	7.91
Upper Quartile	388	2.92	6.68	130.50	19.78
Standard Deviation	80	0.41	1.27	46.63	23.86
González et al., 2000 [50] ^a^			2.6–7.0		
Earley et al., 2006 [22] ^a^		5.1 ± 0.15			
Radostits et al., 2006 [49] ^a^	-	67–175	2.0–7.5	13–21	-
Kaneko et al., 2008 [41] ^a^	0–119	88.4–177	7.14–10.7	17 ± 2	-
Buckham et al., 2008 [21] ^a^	-	-	3.60	13.2 ± 1.3	15.4 ± 1.8
Kataria and Kataria, 2012 [55] ^a^					
Otter, 2013 [51] ^a^	-	44–165	3.4–7.3	13–21	-
Constable et al., 2017 [48] ^a^	-	88–175	2.0–9.6	13–21	
Sánchez et al., 1996 [9] ^b^	36.88	122 ± 16.8	12.8 ± 5.1	165.3 ± 187.6	-
Alonso et al., 1997 [17] ^b^	315	277.6	5.8	-	-
Méndez et al., 2003 [53] ^b^	-	-	16.5	-	-
Bartolomé et al., 2005 [54] ^b^	284	318.2	14.0	-	-
Jordan et al., 2006 [42] ^b^	-	-	5.2–10.3	-	-

^a^ In different breeds. ^b^ Everything in Lidia cattle.

**Table 3 animals-11-02866-t003:** Summary statistics of some enzymes (IU/L) in Lidia bulls’ plasma. Values found by other authors are also shown, both in Lidia breed and in other breeds.

	LDH	CK	AST	ALP	GGT	ALT
n	438	438	438	438	438	438
Mean	2828	6729	495	90	50	59
Geometric Mean	2517	3720	386	84	43	52
Harmonic Mean	2324	2319	321	79	39	48
Sum	1,238,734	2,947,371	216,967	39,223	21,805	25,706
Minimum	1130	167	96	31	11	19
Maximum	24,931	93,641	4476	287	403	384
Lower Quartile	1853	1886	243	67	31	38
Upper Quartile	3120	6838	602	104	55	67
Standard Deviation	1975	10931	462	33	34	35
Radostits et al., 2006 [49] ^a^	692–1445	35–280	78–132	0–500	6.1–17.4	11–40
Kaneko et al., 2008 [41] ^a^	692–1445	4.8–12.1	78–132	0–488	6.1–17.4	11–40
Buckham et al., 2008 [21] ^a^		554.6 ± 68.0				
Kataria and Kataria, 2012 [55] ^a^					21.5 ± 0.24	
Otter, 2013 [51] ^a^	-	-	-	-	0.30	-
Constable et al., 2017 [48] ^a^	692–1445	35–280	78–132	0–200	6.1–17.4	11–40
Purroy (1984, 1985) [56,57] ^b^	3013–4838				21.1	
Sánchez et al., 1996 [20] ^b^	-	532.9 ± 387.5	86.0 ± 15.9	-	-	26.2 ± 3.5
Alonso et al., 1997 [17] ^b^	-	3139.1	318.5	95.5	178.1	49.7
Méndez et al., 2003 [53] ^b^	3483	4286	308	-	-	-
Bartolomé et al., 2003 [54] ^b^	-	1582.8	171.1	-	178.1	77.8

Lactate dehydrogenase (LDH), creatine kinase (CK), aspartate aminotransferase (AST), alkaline phosphatase (ALP), gamma glutamyltransferase (GGT), and alanine aminotransferase (ALT). ^a^ In different breeds. ^b^ Everything in Lidia cattle.

## Data Availability

Not applicable.

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
