# Peer review of "Blood Biochemical Variables Found in Lidia Cattle after Intense Exercise"

_animals, 2021, doi:10.3390/ani11102866_

Round 1
Reviewer 1 Report
The work tittle ‘ Blood biochemical variables found in Lidia Cattle after intense exercise’ has been resubmit to review.
The work is not very novel and limited. Studies about the behavior of fighting bulls in the arena have been showed. The authors have not considered previous suggestions to improve the novelty of the research. It was previously suggested that the different 'encastes' of Lidia's breed be assessed to carry out the research.
References about the phisiological influence of critical effort of animals has been reported (e.g. Spanish horses breeds and other animal species). However, due to the fact that there are few works in this breed with a singular purpose (the fight), the work can provide information to know the peculiarities of the management of this type of "wild" cattle.
This work includes an important bibliographic review about the influence of stress on blood parameters. The authors justify the non-possibility of having experimental data prior to the slaughter of the animals. This circumstance limits the scientific validity of the document. Therefore, the document should be considered as a survey or a bibliographic review document. The paper include 119 references. This number of references is too long for a research paper. Due to the large number of citations and the characteristics of the document presented, the editor should evaluate its presentation as a ‘Bibliographic review article’, despite the fact that the authors present a section with an experimental design.
Introduction:
Line 55: (Rodríguez Montesinos, 2002) is not correct. Author rules are requiered.
During the discussion, the authors refer to the production system of the animals under study. However, the particular production system of these animals is not known. The authors must present the characteristics of the production system of these animals. They are very specifics. Specifically, the fundamentals about the training of animals currently in the field must be presented in the introduction of the document. Author support the discussion in relationship to this argument.
Material and methods:
The description of the animal material is very limited. Some detail of the feeding model must be provided. The authors make references in the discussion about frequent liver problems in these animals.
How many farms practice animal training? At least the percentage with respect to the total of the animals should be known since the work refers to the influence of the training of the animals on tissue resistance.
A few is known about the origin of the animals in this work. I therefore suggest that this document be presented as a bibliographic review or survey.
Results:
Lines 173 and 174. These values (others authors) are in reference to biochemical results after or before ‘fight’? That is not clear.
Lines 158 and 159: It is not possible to refer to an increment of the values since there is no value to compare. The comparison is made with the values of other authors? Perhaps this is the reference to the one made in table 1 (a and b). To make this claim, the authors should have values from biochemical analyzes before the 'fight'.
Lines 158 and 159. The authors cannot say that the values are significantly increased because there is no statistical analysis to support this claim. The authors compare with the results of other authors.
Line 160. Where is the figure 1?
Table 1: Castro et al, 1997[50] don’t have ‘a’ or ‘b’ letter.
In all tables, three, SD: standard deviation, SE: standard error, and CV: coefficient of variation are not necessary. You can choose only one coefficient of dispersion.
Discussion:
Line 237: Values in ranges of other bovine breeds must be provided. Do the authors taken into account the effect of the slaughter age of fighting bulls versus the values of other bovine breeds? Generally, the slaughter ages of other bovine breeds are shorter.
Line 243: Authors must include the results about other cow breeds cited in the text.
Line 260: So ..... do the authors consider that a pathological process is manifested during the fighting process? This statement should be explained.
Lines 272 and 273: So ..... do the authors consider that a pathological process similar to liver degeneration is manifested during the fighting process? This statement should be explained.
Line 279: ‘…..were in range……’ is better than ‘…..similar…..’
Lines 285 and 286: the authors should explain the results. The explanation is not clear. Different authors have shown different results regarding the content of triglycerides in the blood after exercise. The triglyceride content after exercise also depends on the diet plan prior to exercise.
Lines 306 and 307: if the authors make this reference, they should provide information on the population of "amateur" and "professional" bulls, and make a statistical analysis of this situation. The idea is very general to report information.
Line 310: This statement (a single animal) cannot explain the statement.
Lines 314-327: It is not understood what is meant by these three paragraphs. Fighting bulls exercise in the field and does this influence uric acid content after extreme exercise? Or on the contrary they do not and it also affects uric acid levels?
Line 328: ‘Our average creatinine plasma concentration…’ must be changed by ‘Average creatinine plasma concentration observed in Lidia cattle after intense exercise ….’
Lines 340-342 need to be clarified.
Lines 340-341: Are these references about non-stressed animals?
Line 356: Were these values obtained in stressed animals?
Line 360: comparison with human values should not be done. The mechanisms of response to stress can be compared, but the basal levels of human cortisol and glucose should not be compared with the results of the animals of the present study. Basal cortisol and glusose levels in cattle are different from those in humans.
Line 365: How do the speed and duration of exercise directly influence plasma glucose concentrations? Are they going up or down?
Lines 377 to 379: So the study animals have been subjected to stress for more than 20 minutes. This can affect the results.
Lines 394 and 395: Do the authors have an explanation for this observation?
Lines 401. The authors must clarify which animals have carried out a previous training exercise. The results reported by Alonso et al. (17) also performed training exercises. This circumstance must be clarified. The animal training factor seems to affect the results compared to those of other authors. The discussion is confusing. Probably this fact should be presented in the introduction.
Lines 421 to 426 must be eliminated. These lines do not provide important information.
Lines 432. Discussion is not understood. What is the relationship of the activity of GGT, AST ALT with the existence of pre-existing liver disease?
Lines 437 to 439. Are these animals supposed to have liver problems due to overfeeding?
Lines 450-452. Discussion is confusing. Acoording Coppo et al (112) are the increased cortisol values related to a stress response?
Lines 453-455. They were the animals of Alonso et al. [17] and Chaves et al. [103], Bartolomé et al. [52] 454 and González-Buitrago et al [104] stressed?
Lines 462-463. According Sánchez et al. (20). How could the stress of fighting affect your results?
Line 476: What about other groups?
Conclusions:
The conclusions are limited due to the limitation of having data on the profile of the different variables studied before the sacrifice of the animals. The authors should leave this clear limitation in the conclusion section.
Author Response
Attachment is sent with comments for reviewers

Reviewer 2 Report
The study reports biochemical parameters that they associated with intense physical activity, but the study design is based on sampling animals after a painful and stressful death (with a lot of swords and I imagine with some degree of blood loss). I am not aware of the procedures and the slaughter methods during these “fights” and the authors fail to explain in the manuscript, thus It is complicated for me to access if the alterations were from the stressful dead of from the intense exercise.
I am worried about the sampling method since they do not indicate how long after the dead (immediately means, sampling inside the arena).
Another concern is regarding the ethical approval that the authors fail to secure. This paper may result in a huge backlash from the scientific community towards authors and the journal due to animal welfare concerns. I think the approval number would be required here.
I don’t ‘know about Lidia Cattle, thus the authors must include a more detailed description, some herd numbers (importance of this breed), and photos of the animals. The authors mention in the introduction: e. In the year before the fight. This must be explained previously. These cattle are these cattle used for bullfights? (The man with the red cape inside an arena?).
This must be explained in the 1st paragraph to avoid confusion from international readers. I also believed that such animals were brutally killed by swords after each fight, thus, how this would impact the evaluated biochemical variables?
Minor comments. When referring to enzymes they should use activity.
Author Response

(The authors gave the same response as above.)

Round 2
Reviewer 1 Report
The document has been successfully modified.
A suggestion can be made:
Line 163: much greater requires comparison.
Author Response
Dear Editor, Dear Referee,
We greatly appreciate Reviewer 1 for their hard work and dedication. We have tried to respond to your instructions and we have modified the text (in blue).
Thank you.
Referee 1.
Line 163: much greater requires comparison.
Done. We had modified the text to make it clear.
This manuscript is a resubmission of an earlier submission. The following is a list of the peer review reports and author responses from that submission.
Round 1
Reviewer 1 Report
The work tittle “Effect of maximal stress and execcise on biological variables in Lidia Cattle has been submit to review.
The work is not very novel. Studies about the behavior of fighting bulls in the arena have been showed. References in this subject has been reported in horses breeds. However, due to the fact that there are few works in this breed with a singular purpose (the fight), the work can provide important information to know the peculiarities of the management of this type of "wild" cattle.
Only a descriptive analysis of the blood variables is presented in this document. The work could be more interesting if the authors know "the encaste" of the animals. A comparison between "encastes" could be interesting.
I suggest the following:
Tittle:
It is strange that the work is titled Effect of ....... The word "Effect" is important. In the document presented, this objective is difficult to understand since only one measurement value of blood variables related to stress is presented at work. To show an effect, at least one should have values of blood variables before the animals are sacrificed.
Abstract: It is necessary to specify how the samples were obtained from the animals, was it in the sand? Was it at the slaughterhouse? This is important to know because the levels of stress indicators in the blood vary rapidly.
Line 36. It is necessary to specify specifically regarding what value an increase was observed in most of the variables studied.
Keywords: You must include “flightbull”
Introduction:
The introduction is not interesting. It must be rewritten. The document does not present how these animals were raised for their preparation for the Tarurina festival. The purpose of the animals is not clear from the introduction. In fact, it is also not clear from the material and methods of the document.
Lines 45 and 46 are not necessary.
Line 56: requires a reference for Bos Taurus brachiceros.
Lines 64 -65 are not necessary. Bulls are not "athletes"
Line 65. Animals are trained for one purpose: bullfighting festivities.
Line 65: 15 and 20 min. in the breeding or in the bullring? It is not clear.
Lines 70-72 are not necessary.
Line 72. The authors have not explained so far that the animals go to the arena to be slaughtered. And on line 72 they mention the word "the fight". This is a very particular way of slaughtered thess animals must be clarified previously. There is regulation (BOE) in this regard.
Line 80: references to foetal growth are not neccesaries
Lines 83 to 88 are not interesting for the document.
The purpose of the document is unclear. The authors introduce in the objective the hypothesis that short periods of intense activity can suppose a variable response in the indicators of stress. This is hypothesis, it is not the objective.
It is recommended to rewrite the introduction referring to the content of the document.
Material and methods:
The animal material is unclear. The origin of the animals is not evident (line 106: "diverse arena places") The document requires technical data on the number of places where the samples have been collected, the years in which they have been collected.
Does the directive contained in 43 refer to the collection of samples at bullfighting festivals?
Do animals slaughtered at the bullring (arena)?
In the Materials and Methods section we can’t read references about where samples were taken.
Statistical analysis:
The effect of the matador has been contemplated in the study? The effect of the square has been contemplated in the study? Has the effect of the age of the animal been taken into account? Has the effect of the "encaste" of the animal been taken into account?
Results:
Figure 1 and tables 1 and 2 aren’t attached. I can’t see they.
Lines 148 to 150 are discussion. Increase? In reference to?
Line 158. A increase of enzyme proflie was observed. That is results. An increase in enzymatic profile with respect to normal values for the species? What is the normal values for this specie? For dairy cows some values are for beef cows other values.
It would probably have been interesting to have values of blood stress variables in animals at rest (blood samples can be taken during animal sanitation) and compared with blood samples after fighting. The comparison that refers to increment is NOT clear.
I suggest that results be put together with discussion.
Discussion:
Lines 163 to 165 could be eliminated
The discussion does not respond to the results obtained. The authors make references to physiological processes not relevant to the purpose of the document being presented. For example, lines 183 to 198 must be deleted. Effect stress is not show.
Reading the discussion is not easy. The work must present the result of each variable analyzed and then discussed. It is recommended to compare the results with other studies on stress in bovines or even other species such as equine scpecie.
LInes 206 to 208 must be eliminated.
The authors make a description of the results of blood variables related to stress in the case of the bull of Lydia. At this moment I could propose (I cannot see the tables) that the authors present a table with the contributions of other authors with the variables studied. This can help improve work that is based solely on a description. The value of the work is justified by the authors by the high number of samples analyzed from a race for which there is hardly any information. This justifies my proposal. I hope you understand me.
Lines 226 to 233 are good comparisons, but it is required to justify that the stress of transport can also activate stress during Lydia. Remember that transportation is for a long time (usually) and the analysis in this document is the stress response in a short period of time (15 minutes of "lidia").
Lines 244 to 248: We are reference only to a comparative. But, we need discussion.
Lines 265 to 267 are reference to an increase of level choresterol in human from 20 min. In this wok (human) two references were showed the 1st and the 2nd after 20 min. In your paper only one reference to variables from blood stress are reported. That is not comparable.
Lines 280 to 282: What about this reference to human athletes?
A significant documentary effort has been reported by the authors, but most of the time they do not provide relevant information for the document being presented.
The document is written in a bibliographic review format on the determination of stress in animals and humans. It does not respond to the reason for the document.
Conclusions:
The conclusions are acceptable if it is taken into account that the document is a description of results compared with the results of blood variables considered normal for the bovine species. However, they are limited because their own results are compared with results obtained in different situations.
References:
A significant reduction in the number of bibliographic references is required
As a final assessment I suggest that the document be rewritten reducing the content. It is necessary to avoid unnecessary references.
I suggest two possibilities for a new presentation of the document:
- To include a table with all the comparisons of the variables analyzed by other authors and comparation. Because only one measured (only one time after slaughter) of stress has been reported comparisons with other authors is considered a review.
- Another consideration could be made to describe the results. It is suggested to contribute the effect of encaste to know the response of these animals to stress.
The document must include the justification of the results compared to stressful situations. Can not it be a different way.
Reviewer 2 Report
General Comments |
I had a problem reviewing the manuscript as the tables and figures were not included in the document I downloaded. However, from the text, I made the following observations. Although the project used to prepare this manuscript required considerable work, the paper has some inherent problems. First, while the objectives statement describes what was measured, it doesn’t present any hypothesis or reason for measuring these metabolites, hormones and enzymes. What will change as a result of measuring and publishing these results? Second, no blood samples were taken from the bulls prior to the fight or from any type of control animal, there’s no way of saying whether the values changed as a result of the fight or whether high values were just a characteristic of a particular bull. Using values from Lidia cows, horses, and humans for comparison is a weak model. |
Lines |
|
36 and 37 |
One cannot say that the measurements increased as there were no baseline values. |
49 |
Remove the word ‘in’ before dog and camel. |
65 |
The word ‘important’ has no meaning here. Should it be ‘intense’. |
87 |
Omit phrase ‘their blood values’. |
89 |
What is meant by the term ‘correct global body functioning’? |
93 and 95 |
The first time an abbreviation appears in the text, it should be totally spelled out. |
100-103 |
This is was measured, but what was the objective for measuring them? |
107 |
Omit ‘designed’. |
115-138 |
As no measurements were taken before the fight and there was no nonfighting control group, there can be no valid comparison of values. |
144-147 |
The tables and figures are included on the document. |
148 |
What is meant by ‘in terms of metabolic waste’? |
148 and 152 |
Because there are no initial values, the authors can not say that any values were ‘increased’? |
154 |
What is the reference for the ‘normal physiological range for this species’? |
160 |
What is meant by ‘this type of animal’? |
165-169 |
A run-on sentence |
161-199 |
A very long discussion of the effects of various stresses including nutritional, lactation, mixing, transportation, cold, and exercise which are weakly related to fighting bulls without making any comparison to the project. |
202 |
Change ‘that’ to ‘those’ |
206-214 |
Questionable use of other breeds or Lidia cows in comparisons. |
256 |
Can’t say ‘increased’ when there were no baseline values |
259 |
Use of hoses and humans for comparison is weak. |
276-279 |
Unclear sentence. |
283 |
Change to ‘bulls’ |
283-286 |
Unclear sentence. |
322 |
What is meant by the term ‘tame breeds’? |
365 |
Should be ‘1990’s? What is the reference for this statement? |
367 |
Sentence should never begin with an abbreviation. Change to ‘only one of’. |
371 |
Omit ‘the’ |
413 |
Change to ‘similar’. |
422 |
Change to ‘stimuli’. |
423-425 |
In what species are the authors making this comparison and is the term ‘rate’ correctly used here as ‘rate’ implies a change in time? |
453 |
Change to ‘so in bulls with increased’ |